# Evaluation of the Outcome of Local Surgery for Stomal Prolapse

**DOI:** 10.3390/jcm10225438

**Published:** 2021-11-21

**Authors:** Makoto Kosuge, Masahisa Ohkuma, Muneyuki Koyama, Yasunobu Kobayashi, Takafumi Nakano, Yasuhiro Takano, Yuya Shimoyama, Naoki Takada, Tomotaka Kumamoto, Yuta Imaizumi, Hiroshi Sugano, Seiichiro Eto, Yasuhiro Takeda, Saori Yatabe, Ken Eto

**Affiliations:** Department of Surgery, The Jikei University School of Medicine, 3-19-18, Nishi-shimbashi, Minato-ku, Tokyo 105-8461, Japan; masa.ohkuma@jikei.ac.jp (M.O.); takahadoctor@yahoo.co.jp (M.K.); h20mskobayashi@yahoo.co.jp (Y.K.); nakanon_0701@yahoo.co.jp (T.N.); tensai1864@gmail.com (Y.T.); yushimo926@gmail.com (Y.S.); naoki-takada@jikei.ac.jp (N.T.); kumax7356@yahoo.co.jp (T.K.); no.3.flash@gmail.com (Y.I.); hiroshi.sugano@jikei.ac.jp (H.S.); etoh@jikei.ac.jp (S.E.); takeyasu103@yahoo.co.jp (Y.T.); yatabe@jikei.ac.jp (S.Y.); etoken@jikei.ac.jp (K.E.)

**Keywords:** stomal prolapse, stoma reconstruction, laparotomic repair

## Abstract

We reviewed the results of local surgical treatment of stoma prolapse, a long-term complication of stoma construction. Fifteen patients treated for stomal prolapse between 2009 and 2020 at the authors’ and affiliated hospitals were included in this study. The treatment comprised local laparotomic stomal reconstruction (LLSR) in nine patients and stapling repair (SR) in six. We compared and evaluated the clinical and surgical information and postoperative complications. Operation time was significantly shorter in the SR group than in the LLSR group: 20 and 53 min, respectively (*p* = 0.036). The duration of postoperative hospitalization was shorter in the SR group than in the LLSR group: 5.5 and 8 days, respectively; the difference was not significant (*p* = 0.088). No short-term complications were found in either group. Regarding long-term, postoperative complications, parastomal hernias developed after 2.5 years in one patient in the LLSR group and after 6 months in one patient in the SR group; both patients had histories of parastomal hernia surgery and had relatively high body mass indices. Local surgery for stomal prolapse was minimally invasive and performed safely. In patients with a history of surgery for parastomal hernia, attention must be paid to the potential of parastomal hernia developing as a postoperative complication.

## 1. Introduction

Gastrointestinal stoma construction is sometimes necessary in the treatment of gastrointestinal, gynecological, and urinary tract diseases. Approximately 150,000 operations are performed yearly in the USA [1], with artificial anus construction currently the most commonly used technique. However, stoma construction can present problems, with complications reported in 20–70% of cases [2].

Stomal prolapse is a late-stage complication that occurs in 2–26% of cases and involves the inversion and prolapse of the intestine in the stomal region [2]. Once prolapse has occurred, stoma management is more difficult, not only reducing the quality of life but also possibly resulting in hemorrhage or necrosis of the prolapsed intestine, which can sometimes necessitate emergency surgery. Permanent stomas cannot be closed; however, with temporary stomas, radical stomal prolapse treatment involves stomal closure and relocation, which is often difficult in certain cases, such as when a patient is undergoing chemotherapy for advanced and/or recurrent cancer, when there is severe adhesion due to poor general condition, or when several laparotomies have previously been performed. In such situations, there are minimally invasive local surgical methods for *de novo* reconstruction of a stoma at the same locus solely via surgery in the prolapsed stomal region. There have been several reports of such local surgical methods, but no large-scale studies have been performed previously, and the outcome of local surgery for stomal prolapses is unclear. In terms of the principal surgical methods, a few case series have been reported involving the application of Altemeier’s procedure for rectal prolapse [3,4,5] and methods using staplers [6,7,8,9,10,11,12,13,14]. In both methods, the intestine that has prolapsed through the stoma is excised at a level external to the abdominal wall; the stoma is then reconstructed. As no intraperitoneal procedures are performed, these methods offer the advantages of minimal invasiveness and avoidance of intraperitoneal contamination. However, there is still a risk of repeated prolapse if the prolapsed intestine remains and is not completely pulled out; repair of the opened fascial defect region is impossible. In this context, in the institutions where this study was conducted, the technique used was local laparotomic stoma reconstruction, with the exception of stapling repair (SR) as local surgery for stomal prolapse. In this technique of stoma reconstruction, the prolapsed intestine is reliably excised by a minimized laparotomic procedure in the normal region and the excessive fascial defect region is partially plicated.

This study evaluated the results of the aforementioned local surgical treatments for stomal prolapse at the authors’ and other affiliated hospitals.

## 2. Materials and Methods

### 2.1. Patients

The study protocol was approved by the Jikei Institutional Review Board’s Ethics Committee for Biomedical Research (approval no.: 31-474(10056)).

This study included 15 patients who were treated for stomal prolapse between January 2009 and December 2020 at the authors’ hospital or one of the three affiliated institutions. The treatment consisted of a certain type of local surgery, with local laparotomic stomal reconstruction (LLSR) in nine patients and SR in six. Surgery was indicated when there were difficulties with stoma management; worsening of symptoms, such as pain and abdominal distension; or severe edema, hemorrhage, or ischemic necrosis of the prolapsed intestine. Treatment data were obtained from previous medical records, and the following parameters were assessed retrospectively: age, sex, body mass index, level of urgency of previous stoma construction, type of stoma, prolapsed side of loop or double-barreled stoma, prolapsed intestine, length of stomal prolapse, time until prolapse, mode of anesthesia, observation period after reconstruction, operation time, blood loss, length of hospital stay, number of staplers used, and complications.

### 2.2. Local Repair Procedures for Stomal Prolapse

The local surgical methods for treating stomal prolapse were LLSR and SR, depending on the case and the surgeon’s preference. The factors taken into consideration included the patient’s medical condition, treatment status, general condition, and medical history and the condition of the peristomal region. General anesthesia was used for all patients except one (in the SR group), on whom surgery was performed under spinal-epidural anesthesia.

### 2.3. Local Laparotomic Stomal Reconstruction

An incision was made in the mucocutaneous junction of the prolapsed stomal region, and the abdominal and intestinal walls were peeled apart (Figure 1b). Since peristomal intraperitoneal adhesion is often weak, these walls were relatively easy to peel apart. If the loop or double-barreled stoma had only a one-sided prolapse, the abdominal wall and the stoma on the non-prolapsed side were separated. An incision was made into the intestinal wall of the external part of the prolapsed intestine on the antimesenteric side, and the extroverted intestine was reversed (Figure 1c). If any surplus intestine remained within the peritoneum, it was pulled out of the body. If the aperture in the region of the fascial defect had a large diameter, the fascia was plicated by suturing them together on the craniocaudal side using absorbable suture. While paying attention to blood flow in the remaining intestine, the mesentery was treated and the surplus intestine was separated (Figure 1d). Similar to the usual stoma construction mode, the intestinal wall and skin were sutured together using absorbable sutures and the stoma was thus reconstructed de novo (Figure 1e). In the case of a loop or double-barreled stoma, all-layer suturing of the intestinal wall was performed, using an absorbable thread, with the intestine in the stomal region on the non-prolapsed side (Figure 1f).

### 2.4. Stapling Repair

SR was performed according to a previously reported technique [6,9,13]. After anesthesia was administered, the surplus intestine was pulled out as far as possible and allowed to prolapse. Using a linear stapler, either Echelon Linear Cutter (Ethicon Inc., Somerville, NJ, USA) or Endo GIA (Medtronic Inc., Minneapolis, MN, USA), the prolapsed region of the intestine was separated from the tip of the region with the prolapsed intestine to a height of 1 to 2 cm above the skin, in the direction of the mesenteric axis on the mesentery-adhering side, at a 90° angle to the right and the left. The prolapsed region of the intestine was thus divided into two semi-circumferential regions, each of which was separated in the horizontal direction with a linear staple 1 to 2 cm above the skin; the prolapsed region of the intestine was separated. Finally, hemorrhage from the separation line was stopped by performing suturing using absorbable sutures.

### 2.5. Statistical Analysis

Continuous variables were expressed as medians (ranges) and tested using the Mann–Whitney U-test, with *p*-values below 0.05 considered statistically significant. The software used was SPSS^®^ version 22 (IBM, Armonk, New York, NY, USA).

## 3. Results

The patients’ data are presented in Table 1. General anesthesia was used for all except one patient, whose chronic heart failure and chronic obstructive pulmonary disease (COPD) contraindicated the use of general anesthesia. The other patients were administered general anesthesia upon request, for prevention of abdominal pain and discomfort when performing intestinal traction and for muscular relaxation to facilitate fascial plication.

The surgical results for each case are shown in Table 2, and the comparison of the surgical outcomes between LLSR and SR is presented in Table 3. Of the nine patients on whom LLSR was performed, plication of the fascial defect region was done on four patients; three of them had excessive fascial opening size at the time of surgery and one had a parastomal hernia. The surgery duration was significantly shorter in the SR group compared to the LLSR group, at 20 and 53 min, respectively (*p* = 0.036). The amount of bleeding was relatively small (a little to 10 mL) in most of the patients in both groups, but only one patient, with Klippel–Trenaunay–Weber syndrome, in the LLSR group had a large amount of bleeding (850 mL). It is a congenital disorder that causes generalized venous malformations, arteriovenous malformations (fistulas), lymphatic malformations, and capillary malformations. It progresses with growth, is diffusely distributed with indistinct margins across multiple organs, and affects the coagulation system and hemodynamics. In this case, the previous two abdominal surgeries had also resulted in heavy bleeding due to abnormal blood vessel growth and coagulation abnormalities, so the surgical procedure for stoma removal could not be considered to have caused heavy bleeding. The postoperative hospitalization duration tended to be shorter in the SR group than the LLSR group, at 5.5 and 8 days, respectively, but the difference was not significant (*p* = 0.088). In the SR group, the median number of linear stapler cartridges used in the operation was 4, with a range of 2 to 9 stapler cartridges. Postoperative short-term complications were not reported in either group, and no intraperitoneal infection was found in the LLSR group. In terms of long-term complications, parastomal hernias developed 2.5 years after surgery in one patient in the LLSR group and 6 months after surgery in one patient in the SR group. Repair surgery was performed on the patient in the SR group.

## 4. Discussion

Several local surgical techniques for treating stomal prolapse have been reported previously [3,4,5,6,7,8,9,10,11,12,13,14,15,16], but most reports have been case series with no more than 10 patients. The largest number of patients in a previous report was 25, with surgery performed by Koide et al. using staplers [13]; only one patient suffered recurrent stomal prolapse within a year after surgery. However, in an investigation of 16 patients who underwent Altemeier’s procedure, another extraperitoneal local surgical procedure was performed by Mittel et al. [5]; the postoperative stomal prolapse recurrence rate was high (43.6%), with most cases occurring within 1 year. In this study, no short-term complications were found in either group and treatment was feasible with only a short period of postoperative hospitalization. With SR involving intestinal separation only, the operation time was significantly shorter but there was no significant inter-group difference in postoperative hospitalization time. Therefore, the invasiveness due to local laparotomic procedures was not considered to have been high.

With respect to surgical anesthesia, it has previously been reported that SR involves only extraperitoneal procedures and can, therefore, be performed with local anesthesia and intravenous sedation [6,12,14]. In this case series, one patient in the SR group had COPD and chronic heart failure, necessitating surgery under spinal anesthesia, whereas the others received general anesthesia upon request and for prevention of intraoperative abdominal discomfort due to intestinal traction. Often, the aim of LLSR is to achieve plication of the opened fascial defect region in connection with peristomal laparotomy; therefore, surgery can be performed more easily if muscular relaxation is achieved. In principle, either spinal or general anesthesia is considered necessary and all patients who undergo LLSR at the authors’ hospital do so under general anesthesia.

SR, however, has the disadvantage of a high cost, owing to the use of several stapler cartridges. According to a report by Koide et al., [13] the mean number of stapler cartridges used was 4.6 (range: 3 to 8); the median number in this study was 4 (range: 2 to 9). There have been reports on ways of reducing medical costs as much as possible, which involve making holes in the prolapsed intestine at two loci, one at the level of intestinal separation, 1 to 2 cm above the skin, and another passing one fork of the stapler device through those areas to separate the intestine [7,11]. Staplers were first used for all intestinal separations in the authors’ hospital. In recent years, with advancements in surgical devices, when intestinal wall edema is mild, prolapsed intestinal separation in the longitudinal direction is performed using a high-energy device, reducing the number of stapler cartridges used. However, if intestinal wall edema is severe, a stapler device cannot be used. In this study, the surgical method in one patient was switched to LLSR for this reason; thus, care must be taken with its application.

The long-term complications noted in this study included one case of parastomal hernia in each group; in both of these patients, the fascial defect region was not excessively large at the time of the stomal prolapse surgery and no parastomal hernia was found. Therefore, plication of the fascial defect region was not performed at the time of LLSR. Patients with stomal prolapse and parastomal hernia often share numerous risk factors that cause increased intra-abdominal pressure and abdominal wall fragility, including obesity, advanced age, constipation, ascites, and chronic obstructive pulmonary disease [1,17,18]. The factors leading to complications in two patients were a history of parastomal hernia and relatively high body mass indices. In the case of patients with such risk factors, attention must be paid to the potential development of parastomal hernia as a long-term complication after local surgery. However, in the case of the four patients with plication of the fascial defect region performed by LLSR, no parastomal hernia developed. Therefore, plication is considered to have short-term efficacy in preventing stomal prolapse recurrence and parastomal hernia.

### Limitations

The limitation of this study is that it is a retrospective study with a small number of patients, so the results are not conclusive enough to provide clear evidence, as in a randomized trial. Further large studies are needed to clarify the efficacy of local surgery for stoma prolapse compared to stoma closure or reconstruction in different locations in patients with poor general conditions.

## 5. Conclusions

Local surgery for stomal prolapse was performed safely and with minimal invasiveness in patients with poor general health. In the patients with a history of surgery for parastomal hernia, attention must be paid to the potential development of parastomal hernia as a complication after local surgery.

## Figures and Tables

**Figure 1 jcm-10-05438-f001:**
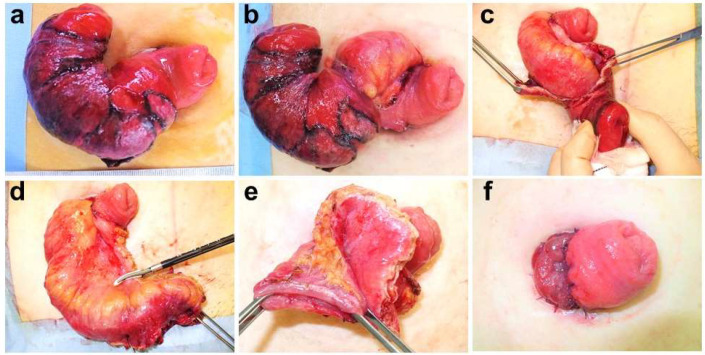
(**a**) Patient with a loop-type ileal artificial anus who presented with anal-side ileal prolapse; (**b**) separation between the intestinal wall and skin and peeling back of the intestine into the peritoneum; (**c**) incision in the extroverted intestinal wall of the prolapsed intestine on the antimesenteric side and reversion to an extroverted state; (**d**) separation of the mesentery of the surplus intestine; (**e**) separation of the surplus intestine and reconstruction of the artificial anus; (**f**) all-layer plication of the intestinal wall between the non-prolapsed side stomal intestines, using absorbable sutures.

**Table 1 jcm-10-05438-t001:** Baseline patient characteristics.

Characteristics	All Patients (*n* = 15)
Age (years)	79 (51–92)
Sex	
Male	9 (60%)
Female	6 (40%)
Body mass index	22.3 (14–27.8)
Urgency of previous operation for stoma construction	
Elective	9 (60%)
Emergency	6 (40%)
Type of stoma	
Loop ileostomy	3 (20%)
Loop colostomy	8 (53.3%)
End colostomy	3 (20%)
Double-barreled stoma of ileum and colon	1 (6.7%)
Prolapse side of loop or double-barreled stoma	
Oral	6 (50%)
Anal	5 (41.7%)
Both	1 (8.3%)
Prolapse intestine (including a both-side case)	
Colon	13 (81.3%)
Ileum	3 (8.7%)
Length of stoma prolapse (cm)	14.5 (5–20)
Duration until prolapse (days)	103 (7–1854)
Anesthesia used	
General	14 (93.3%)
Epidural + spinal	1 (6.7%)
Observation period after reconstruction (days)	193 (11–2069)

The data are presented as a median (range) or as *n* (%).

**Table 2 jcm-10-05438-t002:** The surgical results for each case.

No.	Length of Stoma Prolapse (cm)	Prolapse Intestine	Anesthesia	Surgical Methods	Operation Time (min)	Blood Loss	Length of Hospital Stay (Days)	Postoperative Complications
1	20	Ileum	General	LLSR	55	850 mL	33	None
2	14	Colon	General	SR	20	A little	6	Parastomal hernia
3	12	Colon	General	SR	67	10 mL	5	None
4	20	Ileum	General	LLSR	35	A little	8	Parastomal hernia
5	10	Colon	General	LLSR	61	A little	14	None
6	10	Colon	General	LLSR	60	A little	10	None
7	18	Colon	General	LLSR	46	A little	8	None
8	8	Colon	General	LLSR	70	A little	7	None
9	15	Ileum	General	LLSR	58	A little	6	None
10	15	Colon	General	SR	20	A little	8	None
11	16	Colon	General	SR	15	A little	5	None
12	5	Colon	General	SR	15	A little	2	None
13	Unknown	Colon	Epidural + spinal	SR→LLSR	50	5 mL	10	None
14	Unknown	Colon	General	SR	31	5 mL	18	None
15	Unknown	Colon	General	LLSR	30	A little	8	None

**Table 3 jcm-10-05438-t003:** Comparison of the surgical outcomes between LLSR and SR.

	LLSR (*n* = 9)	SR (*n* = 6)	*p*-Value
Operation time (min)	53 (30–70)	20 (15–67)	0.036
Blood loss (mL)	A little (A little-850 mL)	A little (A little-10 mL)	0.776
Length of hospital stay (days)	8 (6–33)	5.5 (2–18)	0.088
Number of staplers used	-	4 (2–9)	
Complications			
Early	None	None	
Late	Parastomal hernia 1	Parastomal hernia 1	

The data are presented as the median (range).

## Data Availability

The data presented in this study are available from the corresponding author upon reasonable requests. The data are not publicly available since they are covered by institutional privacy policies.

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
