# Peer review of "Evaluation of the Outcome of Local Surgery for Stomal Prolapse"

_jcm, 2021, doi:10.3390/jcm10225438_

Round 1

Reviewer 1 Report

This case series presents the authors´experience with management stoma prolapse. The techniques employed by the authors are well described. The images provided could better with a higher resolution.

Carrying out statistical analysis on this small population makes little sense. The authors should simply describe their study population using case numbers.  

The authors, in my opinion, need to rephrase the passage with 850ml blood loss. Blaming this on "abnormal blood vessels" is wrong. 

This manuscript is primarily about stoma prolapse. However, reading through the manuscript, the reader may get the impression that parastomal hernia is a central aspect of this paper. The authors should focus on stoma prolapse.

Reviewer 2 Report

  • This paper is a description of cases only and not a veery much needed comparative randomized study. A case report is always of interest to operating surgeons. The results of this study are not of any aid for an operating surgeon
  • The selection of patients to the two groups studied is by surgeons’ choice. This makes the results irrelevant and of no scientific interest.
  • The staple method is not thoroughly described while the other method is described by using text supported by pictures.
  • The discussion does not start by stating the study results,
  • The conclusion is not supported by the significant results of the study

Round 2

Reviewer 2 Report

The authors have improved their report.